# Silver linings of the COVID-19 lockdown in New Zealand

**Matthew Jenkins****[1]*, Janet Hoek[2], Gabrielle Jenkin[1], Philip Gendall[2], James Stanley[2], Ben Beaglehole[3], Caroline Bell[3], Charlene Rapsey[4], Susanna Every-Palmer**[1]

1 Department of Psychological Medicine, University of Otago, Wellington, New Zealand, 2 Department of Public Health, University of Otago, Wellington, New Zealand, 3 Department of Psychological Medicine, University of Otago, Christchurch, New Zealand, 4 Department of Psychological Medicine, University of Otago, Dunedin, New Zealand

* matthew.jenkins@otago.ac.nz

## Abstract

The COVID-19 pandemic has caused significant disruption, distress, and loss of life around the world. While negative health, economic, and social consequences are being extensively studied, there has been less research on the resilience and post-traumatic growth that people show in the face of adversity. We investigated New Zealanders' experiences of benefit-finding during the COVID-19 pandemic and analysed qualitative responses to a survey examining mental well-being during the New Zealand lockdown. A total of 1175 of 2010 eligible participants responded to an open-ended question probing 'silver linings' (i.e., positive aspects) they may have experienced during this period. We analysed these qualitative responses using a thematic analysis approach. Two thirds of participants identified silver linings from the lockdown and we developed two overarching themes: Surviving (coping well, meeting basic needs, and maintaining health) and thriving (self-development, reflection, and growth). Assessing positive as well as negative consequences of the pandemic provides more nuanced insights into the impact that New Zealand's response had on mental well-being.

## Introduction

On 30 January 2020, the World Health Organization designated the COVID-19 outbreak a 'public health emergency of international concern' [1]. In response to the pandemic, many governments enacted various states of lockdown and ordered non-essential workers to stay at home. Both the pandemic and the public health measures aimed at containing it have impacted on mental health and well-being, with increased psychological distress, anxiety, and depressive symptoms reported in New Zealand [2] and internationally [3].

Yet, not all psychological outcomes of potentially traumatic events are negative; researchers have also documented post-traumatic *growth* following adversity [4,5]. Post-traumatic growth is the positive psychological change and improved functioning that can occur following adverse events [6]. Thus, in contrast to resilience, which can be conceptualised as 'bouncing back' from a stressful event, post-traumatic growth has been described as 'bouncing forward'

**Data Availability Statement:** All data files are available from the figshare database (DOI:10.6084/m9.figshare.13023065).

**Funding:** The authors received no specific funding for this work.

**Competing interests:** The authors have declared that no competing interests exist.

[7], representing an opportunity to improve aspects of one's life rather than resolve to go back to the way things were pre-event. Tedeschi and Calhoun described main factors that are indicative of growth following exposure to stressful events: relating to others, new possibilities, personal strength, spiritual change, and appreciation of life [8]. Post-traumatic growth has been associated with increased quality, richness and appreciation of life, altered priorities, more meaningful relationships (familial, community), increased personal strength, increased resilience, and improved self-efficacy [6,9–11]. Importantly, positive adjustment may also offset the negative psychological impact of stress-inducing events [11].

Although it is currently too early to understand the long-term psychological effects of COVID-19, post-traumatic growth has previously been shown to occur following disease outbreaks. For example, the SARS virus, while resulting in high levels of psychological distress, also resulted some affected people experiencing positive outcomes such as personal, social and spiritual growth [12]. Early research in the context of COVID-19 has suggested increased post-traumatic growth in healthcare professionals [7]. Consequently, there have been calls to investigate the positive, as well as negative, psychological outcomes of COVID-19 [13].

## Growth via adaptive coping strategies: Searching for silver linings

In response to stressful events, individuals tend to adopt one of several types of coping strategies. For example, emotion-focused coping is aimed toward changing one's emotional response to a stressor (e.g., through distancing or distraction), while those using a problem-focused approach will attempt to alter the stressful situation. A third type of coping strategy is meaning-focused coping, which involves the reappraisal of a stressful event to ascribe it positive meanings [14]. In doing so, people reshape thoughts about the stressor and its role in their life, and generate positive emotions providing needed motivation to sustain coping over the long term.

Finding benefits–or 'silver linings'–under adversity, is one type of meaning-focused coping. Specifically, it refers to the act of finding positive ways that one's life has changed as a result of a traumatic or stressful event [14]. While it can happen alongside emotion-focused and problem-focused coping, it is often utilised when such strategies are not viable, for example when stressors cannot be easily overcome with short-term problem solving or emotion regulation [15].

Benefit finding has the capacity to reduce the psychological impact of stressful events and lead to growth following adversity [16]. For example, people who tend towards positive reappraisal report fewer depressive symptoms when under significant stress [16] and show greater post-traumatic growth following disasters [17]. With regards to coping with COVID-19, research into benefit finding in response to pandemics is growing. As pointed out by researchers [17], a greater understanding of benefit finding in the midst of such stressful events is an important research endeavour. For example, August & Dapzewicz identified benefit finding as a key coping strategy for university students during COVID-19 in the USA, and this was associated with the reduced likelihood of experiencing fear, anxiety, and stress [18]. Similarly, a German study demonstrated a positive association between benefit finding and life satisfaction in the early stages of the pandemic [19].

The type of silver linings may also matter. Enhanced community spirit and social connection are often reported following potentially traumatic events [20,21] and have protective effects on mental health and future resilience [22–24]. Indeed, during the COVID-19 pandemic, a sense of social belongingness was associated with greater psychological well-being in students in Turkey [25]. The threat to social connection posed by COVID-19 was highlighted early on in the pandemic [26], as in-person social connection with people outside residents'

own households was strongly discouraged in many countries. Lockdowns require people to develop alternative ways of maintaining social connections and tapping into community spirit, and pose a unique challenge to whether and how silver linings may be experienced.

### The current study

Previous research suggests that the presence of silver linings in the face of potentially traumatic events is linked to post-traumatic growth and reduced adverse psychological effects. The current study aimed to investigate silver linings reported by New Zealand residents during the COVID-19 lockdown: Identifying such silver linings may inform the support required during prolonged stressful events such as pandemics and associated lockdowns [27].

### Research question

What types of 'silver linings' did participants identify during the COVID-19 lockdown?

## Methodology

### Context and design

During the Level 4 lockdown (from 25 March to 23 April 2020), the New Zealand Government ordered people to stay at home unless they were undertaking essential employment (healthcare workers, providers of essential goods and services), shopping for essential items, or exercising. People were required to refrain from contact with those outside their household (colloquially known as their 'bubble') [28]. This lockdown was widely regarded as one of the strictest in the world [29].

We undertook a large quantitative online survey to investigate the psychological responses of New Zealanders to the COVID-19 Level 4 lockdown [2], which was fielded between 15 and 18 April 2020 (Level 4 lockdown days 19 to 22). In this article we report on a qualitative analysis of comments made in response to an open-ended question within this larger study. While not a typical qualitative research design, our approach mirrors that undertaken by other health researchers who have collected qualitative data as part of a wider study [30].

### Participants and recruitment

Sampling criteria followed those of the parent study, details of which are provided elsewhere [2]. Recruitment occurred via Dynata, a commercial panel survey provider. We applied quotas to ensure our sample approximated the New Zealand population with respect to gender, age, and ethnicity. The survey was fielded using the Qualtrics survey platform. All participants gave consent prior to undertaking the survey; ethical approval was granted by the University of Otago Ethics Committee (F20/003).

### Survey instrument

Participants were asked 'Have you experienced any silver linings or positive aspects during the COVID-19 Level 4 lockdown?' and could respond 'Yes, for me personally', 'Yes, for wider society', or 'No'. Multiple responses were allowed (i.e., both for personal and wider society). Those responding 'yes' to either personal or wider society silver linings were asked: 'What are these silver linings, for you personally or for wider society?', with responses recorded using free text.

Of the 2,010 participants who completed the online survey, 1,227 (64%) reported experiencing silver linings personally ($n = 899$, 45% of the total sample) and/or for wider society ($n = 757$, 38% of the total sample). Fifty-two of these participants did not record an interpretable open text response, giving 1175 usable free text responses.

## Analysis

Coding of open-text output followed a six-stage process of thematic analysis [31]. The six stages involved: 1) familiarisation with the data, 2) generating codes, 3) generating initial themes, 4) reviewing these themes, 5) defining and naming themes, and 6) writing a narrative to synthesise data and integrate it into existing literature. A realist epistemological position was taken, assuming a direct relationship between language, meaning, events, and implications for individuals [31]. We developed themes inductively and did not use *a priori* categories.

We began by analysing the data semantically using a surface reading to identify explicit meanings, but without detailed interpretation or expansion of these. Responses that pertained to multiple themes were divided, with the original intact quote kept in a raw data file. Fiften responses lacked enough context or information to assign to a theme and were categorised as 'unassigned'.

Three researchers (MJ, JH, GJ) undertook the analysis. MJ read the first 500 responses and manually coded each response, focusing on recurring ideas within the text and proposing preliminary themes. Where appropriate, individual quotes that pertained to several codes were divided into smaller meaningful units corresponding to each coded theme (with full intact quotes retained as raw data). JH and GJ read the coded data, compared these to preliminary themes, and identified discrepancies in codes and themes for further discussion. We then developed a thematic map to represent theme clusters, including higher order and lower order themes, which we illustrated with representative quotes. The research team then read the remaining 727 responses and confirmed their allocation to a theme.

## Results

Table 1 shows the demographic characteristics of those respondents who provided an interpretable open text response (*n* = 1175). The median age of respondents was 45 years.

We identified two overarching themes: surviving and thriving, and two cross-cutting themes: social cohesion and perceived agency. Further, specific silver linings were identified as being facilitated by technology and increased time. Table 2 provides a summary of these themes and the relative prevalence of each main theme (percentages in brackets). We note that while we present the prevalence rates of the specific main themes, prevalence rates do not necessarily reflect the relative importance of each theme. Fig 1 depicts the interactions between themes.

**Table 1. Sample demographic characteristics.**

| Demographic | *n = 1175* | % |
|---|---:|---:|
| **Gender** | | |
| Male | 487 | 41.4 |
| Female | 684 | 58.2 |
| Gender diverse | 4 | 0.3 |
| **Ethnicity** | | |
| New Zealand European/Other | 806 | 68.6 |
| Māori | 243 | 20.7 |
| Pacific | 47 | 4 |
| Asian | 79 | 6.7 |
| **Employment status** | | |
| Non-essential employment | 490 | 41.7 |
| Essential employment | 235 | 20 |
| Not in workforce | 450 | 38.3 |

**Table 2. Summary table of themes and facilitators.**

| Theme 1: Surviving | Quote |
| --- | --- |
| **New Zealand is keeping safe** (6% of respondents) | 'I believe the level 4 lockdown has helped the country keep the virus under control.' |
| **Community and caring** (6% of respondents) | 'More effort to help vulnerable members of the community.' |
| **Doing our bit** (3% of respondents) | 'People are following the guidelines.' |
| **Employment and financial security** (2% of respondents) | 'I'm on a temporary contract and due to Covid-19 I have been offered a permanent position giving me job security.' |
| **Theme 2: Thriving** | **Quote** |
| **Societal re-recreation** (50% of respondents) | |
| Increased time with family and friends | 'Getting to spend more time with my kids and spouse.' |
| Improved interactions amongst local communities | 'We have also had some really great communication with our next-door neighbours, who we previously knew but not well' |
| Respite from social commitments | Free time, don't have to socialise.' |
| **Personal reflection and re-creation** (50% of respondents) | |
| Reduced materialism | 'Only shopping for essentials not wasting money on stuff one wants rather than what one needs.' |
| Work re-creation | 'Learnt different ways to work.' |
| Personal development and self-care | 'Personal development', 'I am able to exercise more'. |
| **Environmental re-creation** (27% of respondents) | |
| Personal environment | 'Quieter environment.' |
| Wider environment | '[There is] less air pollution and nature having a break from humans destroying it.' |
| **Cross-cutting themes** | |
| Social cohesion 58% of respondents) | 'We may be a small country but we are doing an amazing job.' |
| Perceived agency (31% of respondents) | 'I can stay at home and pursue the hobbies I enjoy.' |
| **Facilitators of silver linings** | |
| Time (20% of respondents) | 'Time to do some things that never get done at home.' |
| Technology (8% of respondents) | 'Have been using FaceTime with members of my family which I hadn't done before.' |

## Theme 1: Surviving

We conceptualised surviving as 'continuing to live or exist, typically in spite of an accident, ordeal, or difficult circumstances' [32]. This theme incorporates coping, remaining healthy and safe, and maintaining employment and financial security in the face of adversity.

**New Zealand is keeping safe.** Some participants reported silver linings that related to the containment and then elimination of COVID-19 (e.g., 'New Zealand is keeping safe and not increasing cases'; 'We may be a small country but we are doing an amazing job, led by our Prime Minister and our Director General of Health. . . They give us confidence we will get through this').

For others, the silver lining lay in the possibility of learning from the situation, so the country could deal more effectively with future pandemics or similar situations (e.g., 'COVID-19 will help with other viruses in the future as it will develop future prevention').

**Doing our bit.** Many participants reported pride in New Zealanders' willingness to comply with government restrictions by making individual sacrifices to achieve a collective goal ('People are following the guidelines'; 'Kiwis can be well behaved and abide by rules when need be'). Such compliance reflected more than individual responsibility and indicated social

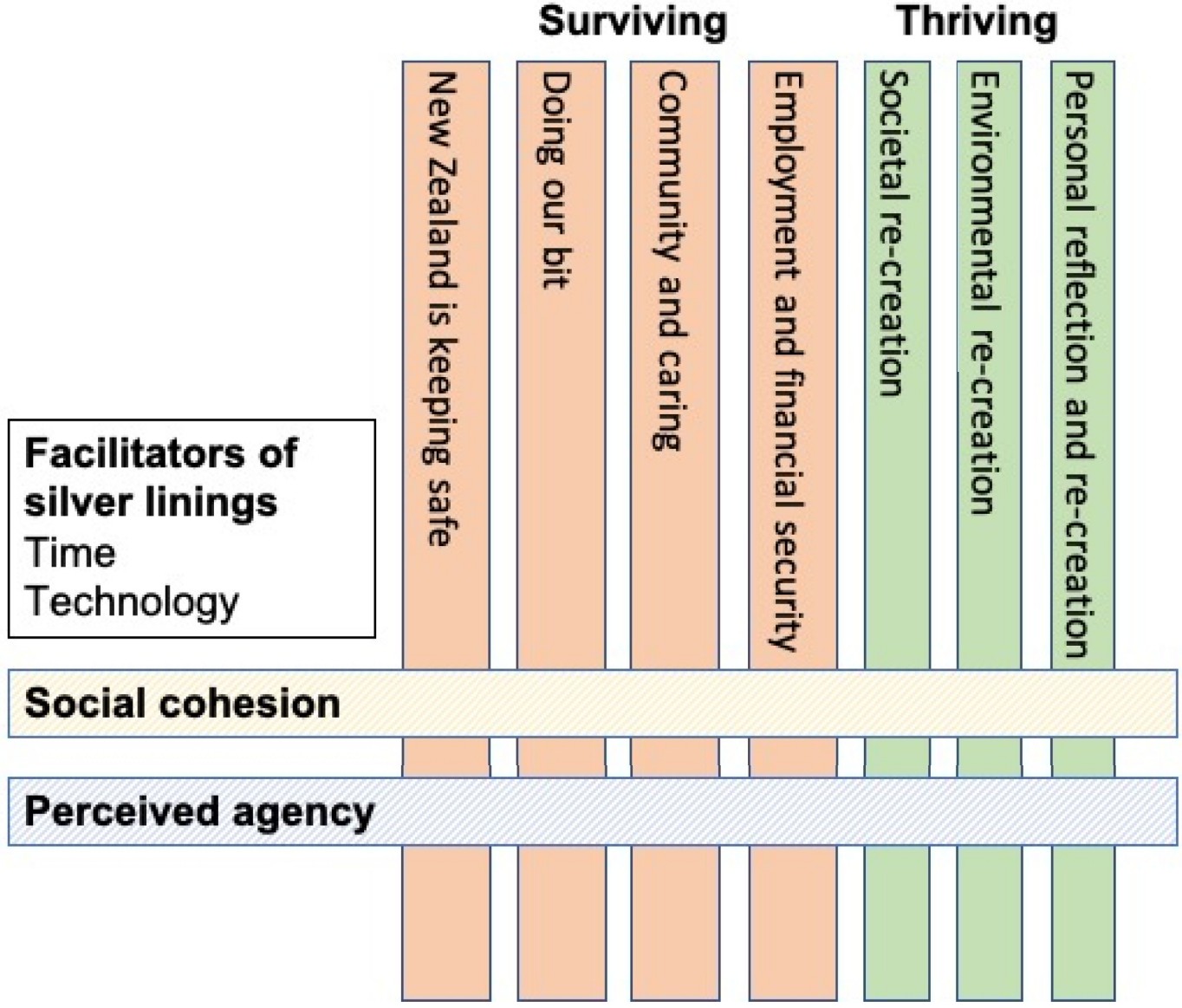

**Fig 1. Model depicting overarching and cross-cutting themes.**

cohesion and commitment. At a macro level, participants' comments indicated increased national unity: 'I think it's brought New Zealanders together, united in our shared COVID experience' and, 'I saw how the people of New Zealand worked together to minimize the impact of COVID-19'. This togetherness suggests greater social cohesion, a theme we explore in more detail below.

**Community and caring.**   Many people reported that kindness and helping behaviours became more common over this period. They described 'an old fashioned sense of community and caring amongst many. . . that was not apparent before lockdown' and that 'neighbours [are] supporting one another', for example 'offers from neighbours and family to do shopping' and 'being kinder to neighbours, grateful comments from the elderly who thought they had been forgotten and were lonely.' As one participant put it: 'People have become more attentive

to each other in society'. Some people found faith-based communities provided social and practical support, including 'regular telephone calls from church friends' or participation in online religious services. Participants also felt greater care had been extended to vulnerable community members (e.g., 'Everyone is more caring of one another especially the less fortunate').

**Employment and financial security.** Some participants saw employment or financial security as their silver linings, and several acknowledged the government's provision of wage subsidies: '[We are] lucky to have a government in place that looks after people with wage subsidies and help during a difficult time'. A small number reported continued or extended employment, which fostered job security (e.g., 'I'm on a temporary contract and due to COVID-19 I have been offered a permanent position giving me job security').

## Theme 2: Thriving

Many participants reported silver linings extending beyond survival, and discussed the lockdown as an opportunity to thrive and grow, often through a process of reflection and re-creation. These participants actively improved aspects of their lives by changing daily habits, reforming relationships, or extending their social networks in new ways. Participants had reflected on their priorities and values, and made changes to their lives. At meso (local community) and macro (national) levels, participants re-connected with their communities, which fostered societal re-creation and increased social cohesion.

**Societal re-creation.** Participants commented on how they had improved interpersonal relationships, which for some was as simple as having more time and opportunity to connect with family members (e.g., 'Spending unlimited or unrestricted time with my child'). Some also reported closer relationships with family members (e.g., 'My relationship with partner and family improved'), and more frequent interactions: 'It has made people interact with their closest family and friends, usually this would only happen on special occasions'. Several felt pleasure at observing other families spending time together, and viewed others' relationships as their silver lining (e.g., 'Seeing families out together walking, playing').

Catalysed by the physical distancing rules in force, technology became a major facilitator of familial and social connections via online services such as videoconferencing. For example, one participant reported how technology meant that they 'got to speak with my dad, who lives overseas, daily', while others described having fun with friends on social media. For example, a participant described joining 'a global online knitting group' to maintain social interactions.

Several participants had re-created or developed relationships with people in their communities: 'We have also had some really great communication with our next-door neighbours, who we previously knew but not well'. Others made completely new connections: 'The interaction of people who don't even know each other is amazing'. While observing social distancing rules, people reported speaking over their fences to their neighbours, 'people smiling and greeting each other', and 'on social media there are always call outs from people who are offering to help families in need with food, and there are people who are just doing spontaneously nice things for others that you didn't notice before.'

In contrast to silver linings associated with increased social connection during lockdown, others have thrived because they had fewer social commitments over this period: 'Free time, [I] don't have to socialise.' For these participants, the lockdown suited their personalities: 'I am an introvert so actually love staying home' and, in some cases, found that they had 'more space and less social anxiety'.

**Personal reflection and re-creation.** The lockdown prompted people to reflect on their values and re-create aspects of their lives. They reported: 'some good down time and time to think about my future' and 'time to reflect and decide what is really important.' Feelings of

appreciation and gratitude increased (e.g., 'Appreciation for everything. More gratitude for the smaller, meaningful and essential things').

Numerous participants said they realised that 'that material items don't matter'. As one stated: 'I think this break has been good for me, and for society. We do not need to go to shops and malls every day. We do not need to be full-time consumers. Shopping should be something we do sometimes, not all the time!' These reflections sometimes translated into revised spending habits, increased frugality, and reduced materialism: 'Only shopping for essentials, not wasting money on stuff one wants rather than what one needs'. Many had reduced expenditure and had more opportunities to save, as they had fewer costs for transport and 'entertainment' ('not eating and drinking at bars and restaurants or ordering takeaway food').

**Work re-creation.** For some, the lockdown enabled work-related development, including finding new ways of working (e.g., 'I have had the time to do assignments to better my job once this is over') and becoming more creative (e.g., 'This period has allowed me to be creative and add value'; 'working on the business and not in the business'). Just as technology facilitated social and familial connection, it also allowed for flexible work practices, such as working from home. For example, video-conferencing allowed professional activities to continue: 'people learn to work smarter, greater use of technology'. This increased working flexibility was frequently cited as a silver lining (e.g., 'I have been able to work from home successfully') as was reduced commuting time (e.g., 'Not feeling rushed to get to work on time and stressing about being late'). The additional time people had enabled them to spend more time with their family and contributed to the increased time with family and friends sub-theme.

**Personal development and self-care.** More broadly, participants used their increased spare time during lockdown to develop new skills or create new habits. One person explained: 'People [are] discovering new talents or skills about themselves'. Some people improved personal health behaviours by changing their diets (e.g., 'Eating less takeaway rubbish food'), increasing physical activity (e.g., 'more time to exercise'), and sleeping (e.g., 'sleeping better'). For others, the lockdown presented an opportunity to address their general well-being by slowing down, resting and relaxing: 'Slower pace of life, not being so stressed about keeping up'. Many engaged in hobbies, such as cooking for pleasure (e.g., 'I can stay at home and pursue the hobbies I enjoy', 'Time to relax and do some of the creative things I enjoy').

**Environmental re-creation.** Increased spare time enabled participants to improve their personal environments by undertaking house maintenance (e.g., '[I] have had a lot more time to do jobs at home such as painting') or gardening (e.g., 'I have time also to look at my vegetable garden'). The lockdown created a 'quieter environment'. Restrictions on motor vehicle use delivered wider environmental improvements including reduced transport-related pollution (e.g., 'less traffic and congestion'), which in turn gave the environment an opportunity to recover. As one participant stated: '[There is] less air pollution and nature [is] having a break from humans destroying it'. The return of birds and birdsong was also noted by many participants: 'Hearing birds sing', 'Seeing more native birds in our neighbourhood', 'Return of birds to cities'. Several participants hoped that these changes would remain beyond COVID-19: 'Think of how healthier the nation would be with no cars on the road!. . . Hopefully in the long run COVID-19 will prove to be a blessing for mankind'.

## Cross-cutting themes

**Social cohesion.** We define social cohesion as ' the extent of connectedness and solidarity among groups in society' [33] (p.261); this term refers to people's community membership and relationships between community members. We observed this cross-cutting theme within the surviving sub-theme of 'an old fashioned sense of community and caring' and the thriving

sub-themes of 'increased time with family and friends' and 'new and improved interactions amongst local communities'.

**Perceived agency.** We define perceived agency as the 'feeling of control over actions and their consequences' [34] (p.1), which is characterised by enacting personal choice and freedom. Perceived agency was central to many silver linings identified under the overarching themes of surviving and thriving; participants often referred to controlling their own actions and saw themselves as active agents of their silver linings. Under the survival theme, people described how they supported others and framed compliance with government restrictions as a choice rather the the imposition of external control.

Perceived agency also featured strongly within the thriving theme; increased work flexibility gave participants freedom to choose the activities they undertook. Some began new past-times (e.g. 'I can stay at home and pursue the hobbies I enjoy'), such as cooking or creative hobbies, self-care and health behaviours, reaching out to friends and family, and active reflection. Even those who found the lockdown offered respite from social commitments demonstrate how agency replaced social obligation.

## Discussion

The COVID-19 lockdown provided a unique opportunity to investigate people's responses to an ongoing threat that caused significant levels of psychological distress in New Zealand [2]. Despite these adverse outcomes, almost two-thirds of people surveyed identified silver linings resulting from the lockdown.

At a base level, the overarching theme of surviving indicated that participants felt they were coping and remaining safe amidst the pandemic. In the New Zealand context, this theme was identified alongside a comprehensive response to the COVID-19 pandemic that saw the virus eliminated [35]. It is unsurprising, therefore, that maintaining health emerged as a silver lining. The Government's firm and clear approach to the virus was valued by numerous participants. Polls conducted just prior to our study indicated that 84% of New Zealanders supported the Government's response to the pandemic; significantly higher support than reported in other countries (e.g., Great Britain, Germany, Japan) [36]. The New Zealand Government received international praise for their response [37].

Further, as economic modelling predicted negative impacts on global and national economies [38], it is not surprising that people felt grateful for continued employment and financial security. As mentioned previously, the capacity to find these benefits may be associated with reduced psychological distress [15,19], and may have ameliorated distress arising from the extended lockdown. For example, we know that financial insecurity can be a secondary source of distress during or following potentially traumatic events [39], thus having a continuous income would have ameliorated this additional stressor. On the other hand, the theme of thriving typifies post-traumatic growth and showcases how people moved beyond survival and took advantage of opportunities for personal, societal, and environmental re-creation.

The cross-cutting themes of social cohesion and perceived agency featured extensively as part of both surviving and thriving, and have important implications for psychological well-being. Both are closely aligned with basic psychological human needs, as described within self-determination theory [40], a well-established meta-theory of human motivation and flourishing. Perceived agency is closely associated with need for autonomy, defined as 'the need to experience self-direction and personal endorsement in the initiation and regulation of one's behavior' [41]. Similarly, social cohesion aligns with the psychological need for relatedness, including 'establish[ing] close emotional bonds with other people. . . [and] feeling socially connected' [41] p.48).

According to self-determination theory, satisfying these needs results in autonomous–or self-determined–motivations for behaviour. Often, participants saw their behaviours as autonomous rather than externally controlled; this perceived autonomous motivation has positive connotations for behavioural adherence and psychological well-being [42,43]. Typically, providing people with personal choice increases their sense of agency. Although the COVID-19 lockdown restricted movement and reduced choices people could exert, providing a clear rationale for measures that reduced choice may have supported people's perceived agency [41].

This reasoning highlights the crucial role of clear government messaging. The New Zealand government provided daily updates on case numbers, recoveries, and testing; the high transparency received international recognition [44,45]. Self-determination theory suggests this messaging strategy would have increased perceived agency and subsequent compliance, because citizens could understand the lockdown rationale. Further, government messaging focused on the role of each New Zealand resident in containing and eliminating COVID-19. The metaphor of New Zealand as a 'team of five million' (the New Zealand population) and the emphasis on kindness [46] focussed people's attention on the shared goal of eliminating COVID-19 and fostered high compliance with the lockdown restrictions.

Such social cohesion and community may have ameliorated the pandemic's adverse psychological effects, as social support has had a protective effect on mental health following adverse events [47,48], and predicts future resilience [24]. Conversely, separateness is often associated with suffering and negative affect [49,50]. The benefits of increased social cohesion also echo earlier disaster recovery research [51,52]. The pertinent threat of COVID-19 to human connection [24], may explain why so many respondents reported seeking social connections.

## Study strengths and limitations

The study took place during the lockdown period, thus providing us with *in situ* insight, as we captured people's experiences as they occurred, rather than relying on subsequent recall. We also obtained an extensive sample that recorded the views of a broadly demographically representative sample of over 2000 New Zealanders. However, we cannot guarantee that the responses obtained were truly representative of the New Zealand publication. For example, findings may reflect the views of participants who provided more articulate or detailed their responses, and so may have have a disproportionate effect on the analysis.

As responses were taken from a self-completed questionnaire, we could not probe responses to clarify ambiguities or potential contradictions. In-depth interviews could complement our findings and explore the rich themes we have identified. Because the COVID-19 situation was dynamic, and our findings represent data captured within a very specific period, when the measures taken appeared to be eliminating the diease. Our data do not enable us to comment on whether the silver linings we identified would also manifest at different stages of the lockdown, post-lockdown, or if the virus returned to the community. Indeed, previous research has shown that positive outcomes such as social cohesion last for a limited amount of time, referred to as the 'honeymoon phase' following adverse events [52]. Thus, the sustainability of such silver linings remains to be seen.

## Implications

Our analysis of participants' silver lining experiences during the New Zealand COVID-19 lockdown in 2020 revealed widespread and diverse beneficial experiences. Social cohesion figured prominently in participants' silver linings and aligned with the Government's

exhortations to 'Be kind' and the conceptualisation of a 'team of five million'. Strategies that facilitate social cohesion could help maintain well-being during future adverse events. Emphasising agency and providing a clear rationale for restrictions appeared to empower survey participants, and could also inform future strategies.

One third of our participants did not report a silver lining. Interventions that stimulate benefit-finding could support these participants during future threats [53,54]. For example, tasks such as brief writing interventions (i.e., asking particpants to writing down the benefits of a given situation) have shown promise in promoting well-being [55].

### Future research

Future research could utilise quantitative longitudinal studies to address questions of causal links, including investigating whether participants who reported silver linings subsequently experienced increased psychological well-being or decreased psychological distress. More generally, examining whether *specific* silver linings protect against psychological distress experienced by people during adversity could help target interventions and provide more specific support to groups known to experience poorer outcomes.

Future research could also investigate whether silver linings varied by gender, age, ethnicity, or employment status. For example, researchers could investigate whether silver linings reported by participants identifying as Māori and Pasifika (both community-focused cultures), place greater emphasis on friends and family. Similar research could be undertaken in other groups with collectivist cultures. Similarly, the living situation of some participants may have affected the types of silver linings reported, or indeed whether they saw any silver linings. For example, if people living alone were less likely to find silver linings, the intervention could be developed for this group to encourage benefit-finding or increase social connection. Finally, individual qualitative interviews or focus groups would provide the opportunity for richer data and greater contextual understanding.

### Conclusion

The silver linings and themes developed in this study reveal how people survived and thrived under adverse circumstances. The unique circumstances of the COVID-19 pandemic and associated societal restrictions were associated with discovery of silver linings, such as increased time for reflection and personal development. Indeed, the lockdown period represented a major flashpoint in people's lives, and created an opportunity to stop, take stock, reflect, connect, and re-create. Our findings suggest that, in a time of turmoil, unrest, and psychological distress, many people nonetheless found silver linings that met fundamental psychological needs for social connectedness and autonomy. Governments and mental health practitioners should be planning how to meet these needs, should future adverse events occur.

### Acknowledgments

We would like to acknowledge and thank Dynata for their support of this research. Thank you to the NZ Ministry of Health, Ministry of Justice, Department of Statistics, and to Anaru Waa, Emma Sutich, Marcellus Paki, Fiona Mathieson, Giles Newton-Howes, and Elliot Bell for expert advice on survey content and design.

### Author Contributions

**Conceptualization:** Matthew Jenkins, Janet Hoek, Philip Gendall, Susanna Every-Palmer.

**Data curation:** Matthew Jenkins, James Stanley, Susanna Every-Palmer.

**Formal analysis:** Matthew Jenkins, Janet Hoek, Gabrielle Jenkin, Susanna Every-Palmer.

**Investigation:** Matthew Jenkins, Janet Hoek, Philip Gendall, Susanna Every-Palmer.

**Methodology:** Matthew Jenkins, Janet Hoek, Philip Gendall, Ben Beaglehole, Caroline Bell, Charlene Rapsey.

**Project administration:** Susanna Every-Palmer.

**Resources:** Matthew Jenkins, James Stanley, Susanna Every-Palmer.

**Supervision:** Janet Hoek, Philip Gendall, Susanna Every-Palmer.

**Writing – original draft:** Matthew Jenkins, Janet Hoek, Gabrielle Jenkin, Susanna Every-Palmer.

**Writing – review & editing:** Matthew Jenkins, Janet Hoek, Philip Gendall, James Stanley, Ben Beaglehole, Caroline Bell, Charlene Rapsey, Susanna Every-Palmer.

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
