## [Decision Letter · Decision Letter 0]

22 Dec 2020

PONE-D-20-36082

Silver linings of the COVID-19 lockdown in Aotearoa New Zealand

PLOS ONE

Dear Dr. Jenkins,

Thank you for submitting your manuscript to PLOS ONE. After careful consideration, we feel that it has merit but does not fully meet PLOS ONE’s publication criteria as it currently stands. Therefore, we invite you to submit a revised version of the manuscript that addresses the points raised during the review process.

This was an interesting reading. I also conducted a similar study in my country (still under review), so it was interesting to compare results. However, I agree with the Reviewers that modifications are needed. Especially: 

- this study has a strange sampling technique for a qualitative study. One would expect that a qualitative study of this kind would adopt saturation sampling, namely Researchers would collect data while performing first coding phases and stop when new themes cease to emerge. On the contrary it seems Authors here adopted a sampling typical of quantitative, questionnaire-based research. Why was it necessary to collect so many testimonies? Since so many data are available, and coding was performed on all of them, I would expect Authors to report descriptives/percentages of themes across the sample. In other words, since a notable sample is available, readers deserve to know how much a given theme was represented across the sample, which were prevalent or rare etc.

- for the same reason, and in accordance with Reviewers, more information and discussion could be added for table 1 information

- again in accordance with Reviewers, more example/excerpts could be added. To put it simply, I encourage Authors to make full use of the rich data they collected

We look forward to receiving your revised manuscript.

Kind regards,

Stefano Triberti, Ph.D.

Academic Editor

PLOS ONE

Journal Requirements:

'We would like to thank Dynata for their generous support of this research..'

'The authors received no specific funding for this work.'

b. Additionally, because some of your funding information pertains to [commercial funding//patents], we ask you to provide an updated Competing Interests statement, declaring all sources of commercial funding.

In your Competing Interests statement, please confirm that your commercial funding does not alter your adherence to PLOS ONE Editorial policies and criteria by including the following statement: "This does not alter our adherence to PLOS ONE policies on sharing data and materials.” as detailed online in our guide for authors  http://journals.plos.org/plosone/s/competing-interests.  If this statement is not true and your adherence to PLOS policies on sharing data and materials is altered, please explain how.

c. Please include the updated Competing Interests Statement and Funding Statement in your cover letter. We will change the online submission form on your behalf.

3. We note you have included a table to which you do not refer in the text of your manuscript. Please ensure that you refer to Table 2 in your text; if accepted, production will need this reference to link the reader to the Table.

Reviewers' comments:

Reviewer's Responses to Questions

**Comments to the Author**

1. Is the manuscript technically sound, and do the data support the conclusions?

Reviewer #1: Yes

Reviewer #2: Yes

2. Has the statistical analysis been performed appropriately and rigorously? 

Reviewer #1: Yes

Reviewer #2: N/A

3. Have the authors made all data underlying the findings in their manuscript fully available?

Reviewer #1: No

Reviewer #2: Yes

4. Is the manuscript presented in an intelligible fashion and written in standard English?

Reviewer #1: Yes

Reviewer #2: Yes

5. Review Comments to the Author

Reviewer #1: Thanks for the opportunity to read your manuscript. Without disclosing my identity, I'm a New Zealand social scientist living abroad (narrows it down to hundreds of people!) and I appreciated the opportunity to engage with this research about how New Zealanders experienced the COVID-19 Level 4 period.

Overall, I believe this manuscript makes an important contribution. Given the ongoing public health issues in creating a social consensus behind social distancing until vaccine take-up, I hope this will be published.

However, there are a couple of areas that need a little revision.

1) You could make more of the gender findings. It's unclear if we can re-interpret the figures in Table 1 as odds of finding silver-linings or not, but if they can the chance of women finding silver linings are significantly greater than for men : 684/345 = 1.98 (women) compared to 487/415 (men).

This finding is broadly consistent with other research in sociology during the pandemic that there are significant differences in reactions by gender. See, for example, this article from Germany, for which the research was conducted around the same time as your work: https://www.tandfonline.com/doi/full/10.1080/14616696.2020.1808692

2) Related to this point, though less important, it might be useful to add a column to Table 1, displaying the composition of the New Zealand population for the same characteristics.

2a) In the panel for Ethnicity in Table 1, you might spell out New Zealand European.

3) Under Environmental Re-creation, I wonder if you might use some of the quotes that mention birds and bird song to illustrate this theme (by the way, thanks for providing the data, this was an enjoyable part of this review)

4) I think that the theme you identify of agency in the "lockdown" is important, and undercuts the terminology of lockdown. This, to me, is the international significance of this paper. Just the basic findings that 2/3 of people found benefits in the lockdown, and the articulation of people coming together for the public health intervention, is evidence that the intervention succeeded because people could see the benefits. Clearly there was some limited enforcement of the limitations imposed under Level 4, but in large part this was achieved because people voluntarily did it.

Around this point, you could cite and engage with the work on the leadership of Ardern.

McGuire, D., Cunningham, J. E., Reynolds, K., & Matthews-Smith, G. (2020). Beating the virus: an examination of the crisis communication approach taken by New Zealand Prime Minister Jacinda Ardern during the Covid-19 pandemic. Human resource development international, 23(4), 361-379.

Wilson, S. (2020). Pandemic leadership: Lessons from New Zealand’s approach to COVID-19. Leadership, 1742715020929151.

5) Once again, my thanks for providing the responses in a data repository. I think that it is important that before publication a wider range of quotes be used. Several quotes are used multiple to illustrate different themes. While quotes can clearly cross-cut themes, I think it would be stronger to have a range of different voices. The current situation suggests that there is not much depth to the responses, but the data doesn't bear this out: there are other quotes in the dataset that could replace duplicated quotes.

6) I hope that in the final dataset provided, you can include the thematic codes as well as the raw quotes.

Reviewer #2: This article presents a qualitative research on the benefits (silver linings) of COVID-19 lockdown in New Zealand, on

people’s life.

In general I think that the paper and in particular the object of study is weak; authors may refer to positive psychology, not only to the specific area of post-traumatic growth, to improve the conceptualization of silver linings, that is the foundational concept of the study yet is little clarified. This should reflect in more specific research objective and interpretation of data.

The authors should improve the introduction as well, especially enhancing references on the epidemic and psychology (an impressive number of studies were published in the last months, especially on journals such as Personality and Individual Differences, Frontiers in Psychology, and PLOS ONE as well. I believe citations to these works should outnumber older citations on other epidemics or disasters).

At the same time, I think that the concept of post-traumatic growth is not very deepened and could be improved.

Table 1 reports differences between participants who reported silver linings or not. Why are these differences relevant? This is not explored in discussion.

In methods, there is scarce explanation for the questions selected for the survey.

Sampling criteria are not specified, which is important since methodology (coding phases) seem to rule out saturation

According to methods, apparently just one researcher coded the responses, while the others worked on the already coded data. Is this common procedure? Wouldn't it be better to include more coders and report inter-rater agreement?

In my opinion, limitations and future research section could be stronger if authors would refer to other literature on psychological outcomes of the pandemic, and contribute to suggest the investigation of specific relevant psychological constructs and/or behaviors.

Finally, there are some typos in the manuscript. I suggest the authors to re-read the manuscript carefully.

6. PLOS authors have the option to publish the peer review history of their article (what does this mean?). If published, this will include your full peer review and any attached files.

Reviewer #1: **Yes: **Evan Roberts

Reviewer #2: No

---

## [Author Response · Author response to Decision Letter 0]

4 Feb 2021

Responses to reviewers have been provided in full in the attached document, titled 'Response to Reviewers'.

---

## [Decision Letter · Decision Letter 1]

8 Mar 2021

PONE-D-20-36082R1

Silver linings of the COVID-19 lockdown in Aotearoa New Zealand

PLOS ONE

Dear Dr. Jenkins,

Thank you for submitting your manuscript to PLOS ONE. After careful consideration, we feel that it has merit but does not fully meet PLOS ONE’s publication criteria as it currently stands. Therefore, we invite you to submit a revised version of the manuscript that addresses the points raised during the review process.

My apologies for the delay but I needed time to analyze the full manuscript and research again based on Authors' response. 

While the effort in responding to review concerns was appreciable, and Reviewers are satisfied with the changes, I am still not convinced about how the sampling and data reporting aspect is managed in the manuscript. Since data were collected by an online source, demographics were collected, and data were coded within a large quantitative study, it is not clear why Authors are not able to report basic information on prevalence and demographics associated with the emerged themes. Authors report (just) one study as example with a similar approach, ref. 30, which I am not sure could be considered comparable because it has a smaller sample and a more specific topic (switching from smoking to vaping...).

In any case, most of the manual or methodological sources on thematic analysis I have seen say that prevalence of themes across the sample is an important information to report. Otherwise, we as readers are left with the impression that some interesting themes were chosen arbitrarily to present but we have no information on their distribution, internal variety and quantitative significance. Results risk to appear merely anecdotal and we are not sure whether the same information could have been found interviewing 500, 100 or 10 participants only. 

If this is really impossible to do, Authors should at least: 

- include thorough discussion of sampling and data reporting and justify their approach with proper methodological references

- detail this issue more in limitations and implications for future research

Furthermore, I was not able to see the research data on the link Authors provided ("website could not be reached"). Please make sure that data are available for consultation according to PLOS ONE's policies

We look forward to receiving your revised manuscript.

Kind regards,

Stefano Triberti, Ph.D.

Academic Editor

PLOS ONE

Journal Requirements:

Reviewers' comments:

Reviewer's Responses to Questions

**Comments to the Author**

1. If the authors have adequately addressed your comments raised in a previous round of review and you feel that this manuscript is now acceptable for publication, you may indicate that here to bypass the “Comments to the Author” section, enter your conflict of interest statement in the “Confidential to Editor” section, and submit your "Accept" recommendation.

Reviewer #1: All comments have been addressed

2. Is the manuscript technically sound, and do the data support the conclusions?

Reviewer #1: Yes

3. Has the statistical analysis been performed appropriately and rigorously? 

Reviewer #1: Yes

4. Have the authors made all data underlying the findings in their manuscript fully available?

Reviewer #1: No

5. Is the manuscript presented in an intelligible fashion and written in standard English?

Reviewer #1: Yes

6. Review Comments to the Author

Reviewer #1: I would encourage you to provide the file with thematic codings as well as the raw quotes as part of the replication dataset.

7. PLOS authors have the option to publish the peer review history of their article (what does this mean?). If published, this will include your full peer review and any attached files.

Reviewer #1: No

---

## [Author Response · Author response to Decision Letter 1]

10 Mar 2021

Responses are attached in a separate document. Many thanks.

---

## [Editor Report · Decision Letter 2]

23 Mar 2021

Silver linings of the COVID-19 lockdown in New Zealand

PONE-D-20-36082R2

Dear Dr. Jenkins,

We’re pleased to inform you that your manuscript has been judged scientifically suitable for publication and will be formally accepted for publication once it meets all outstanding technical requirements.

Kind regards,

Stefano Triberti, Ph.D.

Academic Editor

PLOS ONE
---

## [Editor Report · Acceptance letter]

25 Mar 2021

PONE-D-20-36082R2 

Silver linings of the COVID-19 lockdown in New Zealand 

Dear Dr. Jenkins:

I'm pleased to inform you that your manuscript has been deemed suitable for publication in PLOS ONE. Congratulations! Your manuscript is now with our production department. 

Kind regards, 

on behalf of

Dr. Stefano Triberti 

Academic Editor

PLOS ONE